# Directional Support Vector Machines

**Diogo Pernes** [1,†] 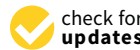**, Kelwin Fernandes** [1,2,†] **and Jaime S. Cardoso** [1,*]

1    INESC TEC and University of Porto, 4200 Porto, Portugal; dpc@inesctec.pt (D.P.); kafc@inesctec.pt (K.F.)
2    NILG.AI, Porto, 4200 Porto, Portugal
*    Correspondence: jaime.cardoso@inesctec.pt
†    These authors contributed equally to this work.

**Abstract:** Several phenomena are represented by directional—angular or periodic—data; from time references on the calendar to geographical coordinates. These values are usually represented as real values restricted to a given range (e.g., $[0, 2\pi)$), hiding the real nature of this information. In order to handle these variables properly in supervised classification tasks, alternatives to the naive Bayes classifier and logistic regression were proposed in the past. In this work, we propose directional-aware support vector machines. We address several realizations of the proposed models, studying their kernelized counterparts and their expressiveness. Finally, we validate the performance of the proposed Support Vector Machines (SVMs) against the directional naive Bayes and directional logistic regression with real data, obtaining competitive results.

**Keywords:** directional statistics; supervised classification; support vector machines

---

## 1. Introduction

Several phenomena and concepts in real-life applications are represented by angular data or, as they are referred to in the literature, directional data. Examples of data that may be regarded as directional include temporal periods (e.g., time of day, week, month, year, etc.), compass directions, dihedral angles in molecules, orientations, rotations, and so on. The application fields include the study of wind direction as analyzed by meteorologists and magnetic fields in rocks studied by geologists.

The fact that zero degrees and 360 degrees are identical angles, so that for example 180 degrees is not a sensible mean of two degrees and 358 degrees, provides one illustration that special methods are required for the analysis of directional data.

Directional data have been traditionally modeled with a wrapped probability density function, like a wrapped normal distribution, wrapped Cauchy distribution, or von Mises circular distribution. Measures of location and spread, like mean and variance, have been conveniently adapted to circular data.

The design of pattern recognition systems fed with directional data has either relied completely on these probabilistic models or just ignored the circular nature of the data.

In this work, we formulate for the first time a non-probabilistic model for directional data classification. We adopt the max-margin principle and the hinge loss, yielding a variant of the support vector machine model.

The theoretical properties of the model analyzed in the paper, together with the robust behavior shown experimentally, reveal the potential of the proposed method.

## 2. State-of-the-Art

Classical methods for method design include probabilistic and non-probabilistic approaches. Probabilistic approaches come in two flavors, generative modeling of the joint distribution $p(x,y)$ and discriminant modeling of the conditional probabilities of the classes given the input. Non-probabilistic approaches directly model the boundaries of the input space or, equivalently, model the partition of the input space in decision regions.

Directional data classifiers have been typically approached [1–3] with generative models based on the von Mises distribution. The von Mises probability density function for the angle $x \in [0, 2\pi)$ is given by:

$$f(x|\mu, \kappa) = \frac{e^{\kappa \cos(x-\mu)}}{2\pi I_0(\kappa)},$$ (1)

where $I_0$ is the modified Bessel function of order zero, $\kappa > 0$ is the concentration parameter, and $\mu$ the mean angle.

Analyzing the posterior probability of the classes,

$$
\begin{aligned}
p(y = 1|x) &= \frac{p(x|y = 1)p(y = 1)}{p(x|y = 1)p(y = 1) + p(x|y = 0)p(y = 0)} \\
&= \frac{1}{1 + \frac{p(x|y=0)p(y=0)}{p(x|y=1)p(y=1)}},
\end{aligned}
$$ (2)

under the von Mises model for the likelihood, it is trivial to conclude that:

$$p(y = 1|x) = \frac{1}{1 + e^{w_0 + w_1 \sin(x+\theta)}},$$ (3)

where $w_0$, $w_1$, and $\theta$ are functions of the mean and concentrations parameters. Recently [4], a directional logistic regression has been proposed that fits Model Equation (3) directly from data. In there, the multidimensional setting was naturally extended to:

$$p(y = 1|\boldsymbol{x}) = \frac{1}{1 + e^{w_0 + \sum_{i=1}^{D} w_i \sin(x_i + \theta_i)}},$$ (4)

where $x_i \in [0, 2\pi)$ and is the $i$th element in vector $\boldsymbol{x}$.

Noting that:

$$w_i \sin(x_i + \theta_i) = w_{i1} \sin(x_i) + w_{i2} \cos(x_i),$$ (5)

where $w_{i1}$ and $w_{i2}$ are obtained from $w_i$ and $\theta_i$, the directional logistic regression model is favorably written as:

$$p(y = 1|\boldsymbol{x}) = \frac{1}{1 + e^{w_0 + \sum_{i=1}^{D} w_{i1} \sin(x_i) + w_{i2} \cos(x_i)}},$$ (6)

enabling the learning task to be solved with conventional logistic regression, by first applying a feature transformation, where each input feature $x_i$ yields two features, $\cos(x_i)$ and $\sin(x_i)$.

## 3. Support Vector Machine

For an intended output $y \in \{-1, 1\}$ and a classifier score $f(x|w)$, the hinge loss of the prediction $f(x|w)$ is defined as $\mathcal{L}(w|x, y) = \max(1 - f(x|w) \cdot y, 0)$, where $x$ is the input and $w$ is the vector of parameters of the model. The Support Vector Machine (SVM) model solves the problem:

$$\arg\min_{w} \sum_{n=1}^{N} \mathcal{L}(w|x_n, y_n) + \lambda ||w||^2, \tag{7}$$

where $\lambda \geq 0$ is a regularization parameter. In standard $\mathbb{R}^D$ spaces, the model $f(x|w)$ is set to the affine form, $f(x|w) = w_0 + \sum_{i=1}^{D} w_i x_i$, and the previous equation can be equivalently written as:

$$\arg\min_{w} \sum_{n=1}^{N} \max\left(1 - y_n \left(w_0 + \sum_{i=1}^{D} w_i x_{n,i}\right), 0\right) + \lambda \sum_{i=1}^{D} w_i^2. \tag{8}$$

In the trivial unidimensional space, the model boils down to $f(x_1|w_0, w_1) = w_0 + w_1 x_1$, and the partition of the input space is defined by a single threshold; see Figure 1a.

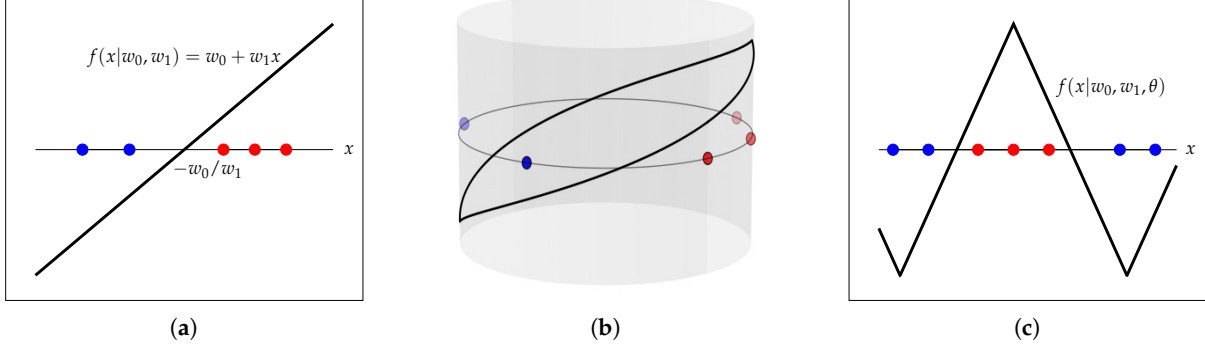

| (a) | (b) | (c) |

**Figure 1.** Toy examples of Support Vector Machine (SVM) in $\mathbb{R}$ and in the unit circle. (**a**) Standard linear model. (**b**) Piecewise linear model in the unit circle. (**c**) Unfolding the piecewise linear model in the unit circle.

In the following, to avoid unnecessarily cluttering the presentation, we will stay in the unidimensional space, returning only in the end to the multi-dimensional problem. We will also assume the period $2\pi$ for the directional data.

## 4. Symmetric Directional SVM

For directional data, the model $f(x_1|w_0, w_1)$ has to be adapted, as it should be periodic, continuous, and naturally take positive and negative values in the circular domain (so it can aim to label positive and negative examples correctly). Arguably, the most natural extension of the linear model in $\mathbb{R}$ is the piecewise linear model in $[0, 2\pi)$; see Figure 1b,c. Note that, now, the partition of the input space requires two thresholds.

Motivated by this observation, we explore models of the form:

$$f(x|w_0, w_1, \theta) = w_0 + w_1 g(x + \theta), \tag{9}$$

where we start by investigating the following specific realizations for $g(x)$:

1.  $g(x) = g_t(x)$, where $g_t(x)$ is the triangle wave with unitary amplitude, period $2\pi$, and maxima at $x = 2k\pi$, $k \in \mathbb{Z}$. This function is piecewise linear, and so, it is close to the linear version in the standard domain.
2.  $g(x) = g_s(x)$, where $g_s(x) = \cos(x)$. This option can be seen as a rough approximation to the intuitive choice $g_t(x)$, but as we will see, analytically more tractable.

### 4.1. Expressiveness of $f(x|w_0, w_1, \theta)$

While in the standard domain, the linear model is able to express (learn) an arbitrary threshold in the input domain, in the directional domain, we need the ability to express any two thresholds. It is easy to conclude that, when instantiated with $g_t$ or $g_s$, the model is able to express two thresholds in the circular domain, whatever their positions are, as formally stated and proven in Proposition 1.

**Proposition 1.** *Let $g : \mathbb{R} \mapsto [-1, 1]$ be an even periodic function with period $2\pi$. For any distinct $x_1$ and $x_2$ in $[0, 2\pi)$, there exists a $\theta$ in $[0, 2\pi)$ such that $f(x) = w_0 + w_1 g(x + \theta)$ is zero at $x = x_1$ and $x = x_2$.*

**Proof.** Set $x_c = 0.5(x_1 + x_2)$ and $\delta = 0.5(x_2 - x_1)$. Note that $x_1 = x_c - \delta$, $x_2 = x_c + \delta$ and $-\pi \le \delta \le \pi$. Now, setting $\theta = 2\pi - x_c$, $w_0 = -g(\delta)$, and $w_1 = 1$ yields a model $f(x) = w_0 + w_1 g(x + \theta)$ with the desired zeros. $\square$

For the $g_s$ option, using Equation (5), $f(x|w_0, w_1, \theta)$ can be equivalently written as $f(x|w_0, w_1, \theta) = w_0 + w_{11}g_s(x) + w_{12}g_s^\perp(x)$, where $g_s^\perp(x) = g_s(x - \pi/2) = \sin(x)$. Therefore, we consider the following equivalent model:

$$f(x|w_0, w_{11}, w_{12}) = w_0 + w_{11}g_s(x) + w_{12}g_s^\perp(x). \tag{10}$$

Similar to the result obtained with the directional logistic regression, the optimization problem in Equation (7) can be efficiently solved by first transforming each directional feature $x$ into two new features, $g_s(x)$ and $g_s^\perp(x)$, and then relying on efficient methods for the conventional primal SVM, such as Pegasos [5].

Unfortunately, the analogous equivalence does not hold for the triangle wave $g_t$. For $g_t$, $f(x|w_0, w_1, \theta)$ cannot be written as $w_0 + w_{11}g_t(x) + w_{12}g_t^\perp(x)$, where $g_t^\perp(x) = g_t(x - \pi/2)$. Still, we could be led to assume the decomposition Equation (10) as a good approximation, when instantiated with $g_t$ and use it in practice, with the benefit of using standard SVM toolboxes in pre-processed data. However, the expressiveness of this model is quite limited. For instance, the model $w_0 + w_{11}g_t(x) + w_{12}g_t^\perp(x_1)$ is unable to learn two thresholds in $[0, \pi/2]$. Since this model is linear in this interval, the result follows.

As such, for $g_t$, we solve the learning task defined by Equation (7) using sub-gradient methods.

## 5. Kernelized Symmetric Directional SVM

By the representer theorem, the optimal $f(x)$ in Equation (8) has the form:

$$f(x) = w_0 + \sum_{n=1}^{N} \alpha_n k(x, x_n), \tag{11}$$

where $k$ is a positive-definite real-valued kernel and $\alpha_n \in \mathbb{R}$. Benefiting from the decomposition of each directional feature in two, this formulation is directly applicable to the primal, fixed margin, directional SVM, when using $g_s$. As such, all the conventional kernels can be applied in this extended space.

When the model is instantiated with $g_s$, $x$ is mapped in a two-dimensional feature vector, $\phi(x) = (\cos(x), \sin(x))^\intercal$, and the inner product between $\phi(x)$ and $\phi(z)$ becomes $\langle \phi(x), \phi(z) \rangle = \cos(x)\cos(z) + \sin(x)\sin(z) = \cos(x - z)$. As such, the feature transformation can be avoided by setting as the kernel the cosine of the angular difference, $k(x, z) = \cos(x - z)$.

As seen before, in the case of $g_t$, a similar conclusion does not hold. However, the result for $g_s$ suggests also investigating the interest of using the function $g_t(x - z)$ as a kernel. We start by presenting Theorem 1, where we show that a broad family of functions, which includes both $g_s$ and $g_t$, may be used to construct formally-valid kernels.

**Theorem 1.** *Let $h : \mathbb{R} \mapsto \mathbb{R}$ be a periodic function with period $T$ and absolutely integrable over one period. Define $g : \mathbb{R} \mapsto \mathbb{R}$ as the autocorrelation of $h$, i.e.:*

$$g(x) = [h \star h](x) = \int_0^T h(t)h(x + t)\, dt. \tag{12}$$

*Then, $k(x, z) = g(x - z)$ is a kernel function, i.e., there exists a mapping $\phi$ from $\mathbb{R}$ to a feature space $\mathbb{X}$ such that $\langle \phi(x), \phi(z) \rangle = g(x - z)$.*

**Proof.** See Appendix A.1. □

**Remark 1.** *The triangle wave $g_t$ is the autocorrelation, as defined in this paper, of a square wave with amplitude $1/\sqrt{2\pi}$ and period $2\pi$.*

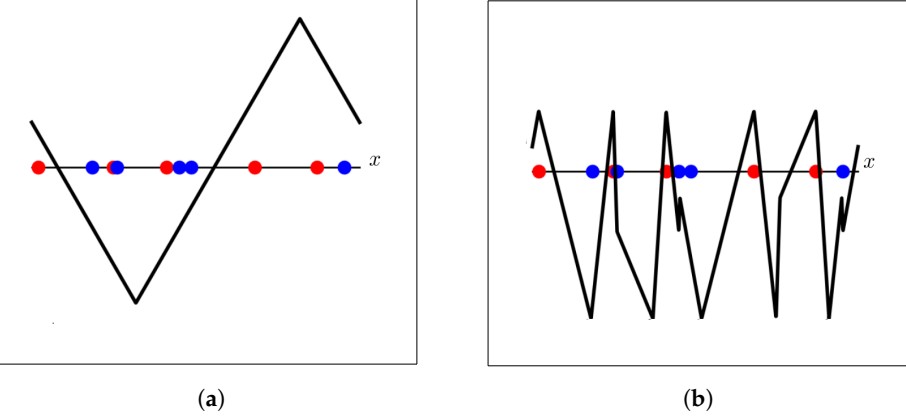

(**a**)　　　　　　　　　　　　　(**b**)

**Figure 2.** Example illustrating the benefits of the triangle wave kernel. (**a**) The SVM with the triangle wave in the primal form cannot learn more than two thresholds; therefore, some training points in this toy dataset are misclassified. (**b**) The SVM with the triangle wave kernel can learn an arbitrary number of thresholds, classifying every training point in this toy dataset correctly.

Having proven the validity of $g_t$ as a kernel, we now focus on investigating the expressiveness of the resulting SVM. The note made in Section 4.1 supports that the sum of triangle functions centered in fixed positions is not expressive enough since it cannot place the decision boundaries in arbitrary positions. However, the kernelized version in Equation (11) can still be appealing, since now, the models are centered in the training observations, and, as such, adapted in number and phase to the training data. For the purpose of this analysis, the notion of the Vapnik–Chervonenkis (VC) dimension [6], given in Definition 1, will be useful.

**Definition 1.** *A parametric binary classifier $l(x|w)$, with parameters $w$, is said to shatter a dataset $x = (x_1, ..., x_N)$ if, for any label assignment $y = (y_1, ..., y_N)$, there exist parameters $w$ such that $l(x|w)$ classifies correctly every data point in $x$. The VC dimension of $l(x|w)$ is the size $N$ of the largest dataset that is shattered by such a classifier.*

Thus, the VC dimension of a classifier provides a measure of its expressive power. In Theorem 2, we establish a result that determines the VC dimension of the kernelized SVMs we are considering.

**Theorem 2.** *Let $k(x,z) = \boldsymbol{\phi}(x)^\mathsf{T}\boldsymbol{\phi}(z) = g(x-z)$ be a kernel function where g is defined as in Theorem 1. Furthermore, suppose that g has zero mean value and its Fourier series has exactly N non-zero coefficients. Then, the VC dimension of the classifier:*

$$l(x|w_0, \boldsymbol{w}_d) = \text{sign}(f(x|w_0, \boldsymbol{w}_d)) = \text{sign}(w_0 + \boldsymbol{w}_d^\mathsf{T}\boldsymbol{\phi}(x)) \tag{13}$$

*equals $2N + 1$.*

**Proof.** See Appendix A.2. □

The Fourier series of the triangle wave $g_t$ has an infinite number of non-zero coefficients, and therefore, the classifier instanced with the triangle wave kernel $k(x,z) = g_t(x-z)$ has infinite VC dimension. On the other hand, the VC dimension of the classifier instanced with the cosine kernel $k(x,z) = \cos(x-z)$ equals three. Consequently, the SVM with the triangle wave kernel is able to express an arbitrary number of thresholds in the circular domain, unlike the SVM with the cosine kernel or with the triangle wave in the primal form, which, as proven before, can only express two thresholds in $[0, 2\pi)$. Figure 2 illustrates these differences.

However, depending on the relative position of the data points, even the SVM with the triangle wave kernel may fail to assign the correct label to all of them. In order to overcome this limitation, composite kernels, constructed from this baseline, can be explored. Typical cases include the polynomial directional kernel, $k(x,z) = (g(x-z)+1)^d$, where $d$ is the polynomial degree and the directional RBF kernel, $k(x,z) = \exp(\kappa g(x-z))$ (while the standard RBF kernel relies on the Gaussian expression, the directional RBF kernel relies on the expression of the von Mises distribution).

Figure 3a shows a simple training set that is correctly learned both with the primal and kernel triangle wave formulations. On the other hand, it should be clear that the setting in Figure 3b cannot be correctly learned by these same models. In this case, setting the SVM with the kernel $k(x,z) = (g_t(x-z)+1)^2$ achieves the correct labeling.

It is important to note that standard, off-the-shelf, toolboxes can be used to solve the kernelized directional SVM directly. One just needs to properly define the kernel as discussed before.

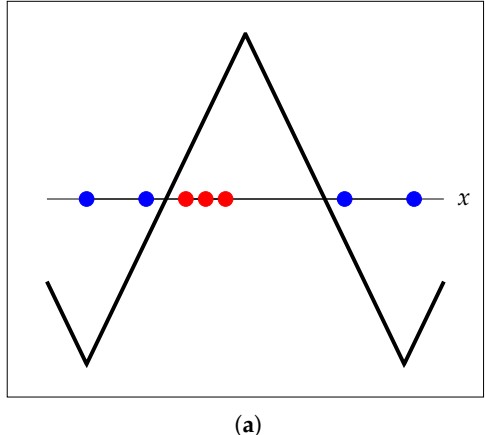
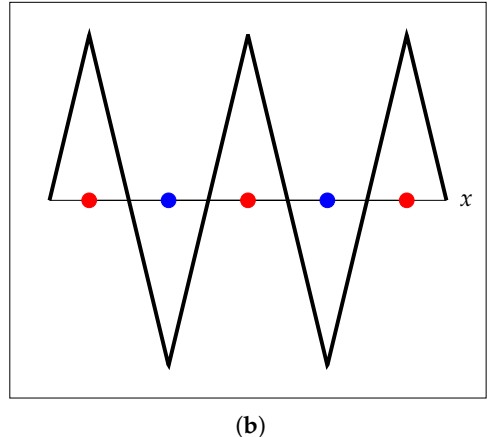

(**a**) (**b**)

**Figure 3.** Example illustrating the benefits of composite kernels. (**a**) Toy example correctly solved by the primal and the kernel formulation (using the directional kernel $g_t$). (**b**) Toy example not solved correctly by the primal, nor the kernel with $g_t$. Solved correctly with polynomial directional kernel of degree two.

## 6. Asymmetric Directional SVM

In Figure 4, we portray a toy dataset together with the model that optimizes Equation (7) using $g_t$ in the model $f(x)$. As observed, the margin is determined by the "worst case" transition between positive and negative examples. It is reasonable to assume that a model placing the second threshold centered in the gap between positive and negative examples would generalize better. Shashua and Levin [7] faced a problem with similar characteristics when addressing ordinal data classification in $\mathbb{R}^D$. Similar to them, we propose to maximize the sum of the margins around the two threshold points. Towards that goal, we only need to generalize the model to allow independent slopes in the two parts of the triangle wave, setting $f(x|w_0, w_1, \boldsymbol{\theta}, \zeta) = w_0 + w_1 g_{asy}(x + \boldsymbol{\theta}, \zeta)$, where:

$$g_{asy}(x, \zeta) = \begin{cases} 1 - \frac{x - 2k\pi}{\pi(1 - \zeta)}, & 2k\pi \leq x \leq 2\pi(1 - \zeta) + 2k\pi \\ \\ 1 + \frac{x - 2k\pi}{\pi\zeta}, & -2\pi\zeta + 2k\pi \leq x \leq 2k\pi \end{cases}, \tag{14}$$

$$\zeta \in (0, 1), k \in \mathbb{Z}.$$

Here, $\zeta$ controls the asymmetry of the wave: if $\zeta \to 0$, the wave has infinite ascending slope; if $\zeta \to 1$, the wave has infinite descending slope, and for $\zeta = 0.5$, it coincides with the symmetric case, $g_t$. The wave is depicted in Figure 5 for some values of $\zeta$.

It should be clear that the model instanced with $g_{asy}$ retains the same expressiveness as before, being able to express two thresholds (and not more than two) in any position in $[0, 2\pi)$.

As before with $g_t$, it is not possible to solve the optimization problem as a conventional setting, and we again directly optimize the goal function using sub-gradient methods.

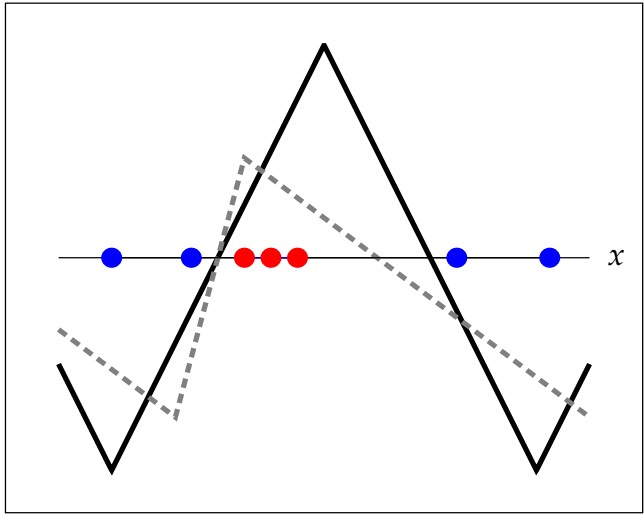

**Figure 4.** Toy example where the fitted symmetric model does not provide an appealing solution.

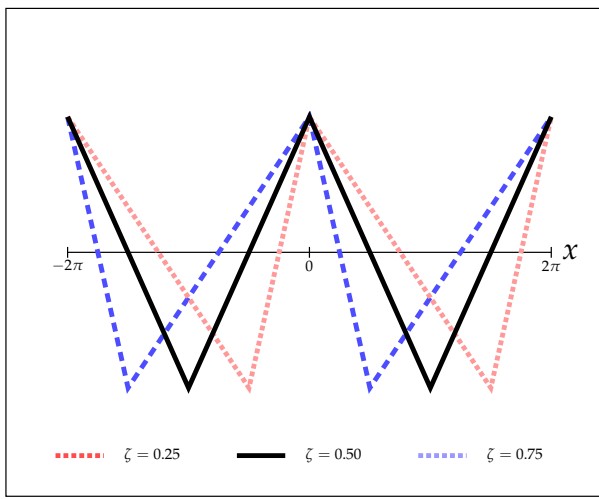

**Figure 5.** Asymmetric model $g_{asy}(x, \zeta)$, for $\zeta \in \{0.25, 0.5, 0.75\}$. This model allows the maximization of the sum of the margins.

## 7. Kernelized Asymmetric Directional SVM

Before, motivated by the behavior with $g_s$ and the representer theorem, we explored models of the form $f(x) = w_0 + \sum_{n=1}^{N} \alpha_n g_t(x - x_n)$. Using the decomposition depicted in Figure 6, $g_t(x) = g_{t_1}(x) + g_{t_2}(x)$, we can rewrite the model as $w_0 + \sum_{n=1}^{N} \alpha_n \left( g_{t_1}(x - x_n) + g_{t_2}(x - x_n) \right)$. We can now gain independence in the two slopes of the model by extending it to $w_0 + \sum_{n=1}^{N} \alpha_{n_1} g_{t_1}(x - x_n) + \alpha_{n_2} g_{t_2}(x - x_n)$, where $\alpha_{n_1}$ and $\alpha_{n_2}$ are two independent parameters to be optimized from the training set.

Since $g_{t_2}(x) = g_{t_1}(-x)$, the model equals:

$$f(x) = w_0 + \sum_{n=1}^{N} \alpha_{n_1} g_{t_1}(x - x_n) + \alpha_{n_2} g_{t_1}(x_n - x). \tag{15}$$

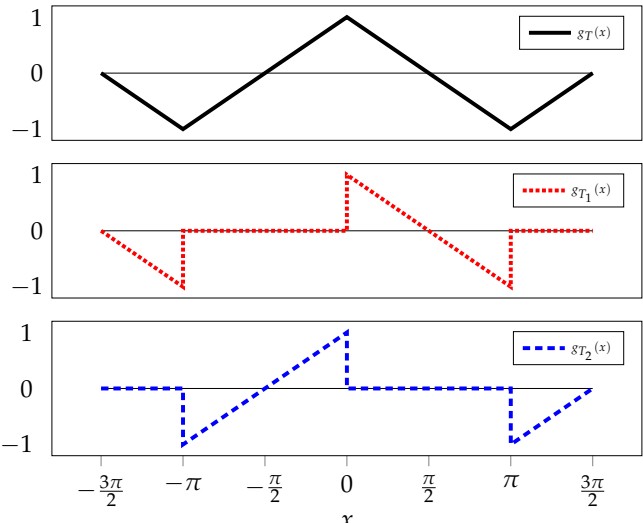

**Figure 6.** Decomposition of the triangle wave.

## 8. The Multi-Dimensional Setting

The extension of the ideas presented before to the multi-dimensional setting is easy. For this purpose, assume our data consist of both directional and non-directional components. This allows each data example to be represented as a vector $x = \left( x^{(d)\top} \quad x^{(l)\top} \right)^\top$, where $x^{(d)} \in \mathbb{R}^D$ represents the directional components and $x^{(l)} \in \mathbb{R}^L$ represents the non-directional ones. Suppose we wish to represent the $i$th directional component $x_i^{(d)}$ in a feature space $\mathbb{X}_i^{(d)}$, through a mapping $\phi_i^{(d)} : \mathbb{R} \mapsto \mathbb{X}_i^{(d)}$, and the non-directional ones in a feature space $\mathbb{X}^{(l)}$, through a mapping $\phi^{(l)} : \mathbb{R}^L \mapsto \mathbb{X}^{(l)}$. Then, our model $f(x|w)$ becomes:

$$
\begin{aligned}
f(x|w) &= w_0 + \sum_{i=1}^{D} w_i^{(d)\top} \phi_i^{(d)} \left( x_i^{(d)} \right) + w^{(l)\top} \phi^{(l)} \left( x^{(l)} \right) \\
&= w_0 + \bar{w}^\top \begin{pmatrix} \phi_1^{(d)} \left( x_1^{(d)} \right) \\ \vdots \\ \phi_D^{(d)} \left( x_D^{(d)} \right) \\ \phi^{(l)} \left( x^{(l)} \right) \end{pmatrix},
\end{aligned}
\tag{16}
$$

where $\bar{w} = \left( w_1^{(d)\top}, \cdots, w_D^{(d)\top}, w^{(l)\top} \right)^\top$. Therefore, in the standard setting where the feature spaces are fixed and possibly infinite dimensional, but the respective inner products have a closed form, we may use the kernel trick to solve the optimization problem. Such kernel is an inner product in the joint feature space $\mathbb{X}_1^{(d)} \times \cdots \times \mathbb{X}_D^{(d)} \times \mathbb{X}^{(l)}$ and equals the sum of the individual kernels:

$$
k(x, z) = \sum_{i=1}^{D} k_i^{(d)} \left( x_i^{(d)}, z_i^{(d)} \right) + k^{(l)} \left( x^{(l)}, z^{(l)} \right),
\tag{17}
$$

where $k_i^{(d)}(\cdot, \cdot) = \langle \phi_i^{(d)}(\cdot), \phi_i^{(d)}(\cdot) \rangle$ and $k^{(l)}(\cdot, \cdot) = \langle \phi^{(l)}(\cdot), \phi^{(l)}(\cdot) \rangle$.

If the feature mappings $\phi_i^{(d)}$ are finite dimensional functions that depend also on parameters to be optimized, like, for instance, in the case of $\phi_i^{(d)} \left( x_i^{(d)} \right) = g_t \left( x_i^{(d)} + \theta_i \right)$, the kernel itself becomes dependent on such parameters. In this setting, we opted to plug Equation (16) directly into Equation (7), solving the problem directly in its primal form using gradient-based optimization.

For simplicity, we set $\phi^{(l)}$ to the identity in our experiments, inducing the usage of the linear kernel for all the non-directional components.

## 9. Experiments

In this section, we detail the experimental evaluation of the proposed directional support vector machines against two state-of-the-art directional classifiers: the von Mises naive Bayes [8] and the directional logistic regression [4]. Following [4], the $\kappa$ parameter of the von Mises distribution was approximated by 100 iterations (a much larger number of iterations than required to have good convergence values) of Newton's method proposed by Sra [9].

The SVM regularization constant $C = \lambda^{-1}$ was chosen using a stratified 3-fold cross-validation strategy. The range of explored values was $10^{-3}, \ldots, 10^3$. The concentration parameter $\kappa$ of the directional RBF kernel was also selected through 3-fold cross-validation in the range $10^{-3}, \ldots, 10^1$. The primal directional SVM with fixed margin was randomly initialized and optimized using Adam [10] for 500 iterations. On the other hand, we initialized the asymmetric primal SVM with the fixed margin margin

parameters after 400 iterations. Then, all the parameters were fine-tuned for an additional 100 iterations. Using pre-trained parameters for the SVM with an asymmetric margin facilitates convergence given the coupled effect of the $\theta$ and $\zeta$ parameters. The kernelized directional version with cosine and triangular kernels was optimized using the standard libsvm [11], which implements an SMO-type algorithm proposed in [12]. For the asymmetric kernel, we used the aforementioned fine-tuning approach in order to fine-tune the $\alpha$ coefficients obtained by the standard toolkit.

We validated the advantages of the proposed approach using 12 publicly-available datasets: Arrhythmia [13], Behavior [14], Characters [15], Colposcopy, Continents, eBay [16], MAGIC [17], Megaspores [18], OnlineNews [19], Temperature1, Temperature2, and Wall [20]. Relevant properties about these datasets (e.g., number of directional and non-directional features, number of classes, dimensionality) are presented in Appendix B. Experiments in previous works [4,8] have shown that directional classifiers outperform traditional ones in these datasets, proving that directionality is an important attribute to exploit. Further details about the datasets, including their acquisition and preprocessing, were presented in [4]. Additionally, in order to facilitate the convergence of the SVM-based models, all the non-directional features were scaled to the range 0–1.

Multiclass instances were handled using a one-versus-one approach for all the binary models (i.e., logistic regression and support vector machines). All the experiments detailed below were executed with a 3-fold stratified cross-validation technique (i.e., by preserving the percentage of samples for each class), selecting the best model in terms of accuracy, and the results of 30 different runs were averaged. Specifically, for each model and dataset, we have evaluated the accuracy and the macro $F_1$-score, which corresponds to the unweighted mean value of the individual $F_1$-scores of each class. Results of these experiments are summarized in Tables 1 and 2, exhibiting average accuracy and macro $F_1$-score, respectively, for 30 independent runs. The best results for each dataset are marked in bold. For reproducibility purposes, the source code and the training-testing partitions are made available (https://github.com/dpernes/dirsvm). The results achieved by the von-Mises naive Bayes (vMNB) and directional Logistic Regression (dLR) align with the results reported in the literature [4].

**Table 1.** Average accuracy and standard deviation per model on 30 runs for 12 datasets, which are presented in increasing order of their respective cardinality. The evaluated models are: Naive Bayes with von Mises distribution (vMNB), directional Logistic Regression (dLR), Multilayer Perceptron (MLP), SVM with directional RBF kernel (dRBF-SVM), SVM with cosine kernel (cos-SVM), primal SVM with triangle wave (t-SVM), SVM with Triangle wave kernel (T-SVM), primal SVM with asymmetric triangle wave (a-SVM), and SVM with Asymmetric triangle wave kernel (A-SVM). Best results are given in bold.

| Dataset | vMNB | dLR | dRBF-SVM | cos-SVM | t-SVM | T-SVM | a-SVM | A-SVM |
|---|---|---|---|---|---|---|---|---|
| **Colposcopy** | $69.56 \pm 11.1$ | $80.56 \pm 9.0$ | $83.52 \pm 8.9$ | $81.67 \pm 7.7$ | $83.00 \pm 9.6$ | $83.48 \pm 8.5$ | $\textbf{85.41} \pm 8.5$ | $82.67 \pm 9.6$ |
| **Behavior** [14] | $46.22 \pm 11.1$ | $82.09 \pm 6.0$ | $80.45 \pm 4.2$ | $80.58 \pm 3.5$ | $\textbf{82.99} \pm 5.5$ | $81.58 \pm 4.0$ | $81.97 \pm 5.3$ | $80.96 \pm 3.8$ |
| **Arrhythmia** [13] | $67.59 \pm 4.7$ | $\textbf{79.06} \pm 6.2$ | $78.51 \pm 4.7$ | $78.59 \pm 5.2$ | $78.51 \pm 6.0$ | $76.49 \pm 6.1$ | $78.82 \pm 5.7$ | $76.41 \pm 5.9$ |
| **eBay** [16] | $83.95 \pm 5.9$ | $85.47 \pm 4.9$ | $\textbf{86.00} \pm 5.1$ | $83.94 \pm 4.6$ | $83.36 \pm 6.2$ | $82.02 \pm 5.7$ | $83.55 \pm 6.4$ | $81.21 \pm 5.3$ |
| **Megaspores** [18] | $76.77 \pm 3.4$ | $\textbf{76.94} \pm 4.2$ | $76.15 \pm 4.4$ | $75.59 \pm 4.7$ | $76.18 \pm 4.1$ | $75.24 \pm 4.2$ | $76.28 \pm 3.7$ | $75.69 \pm 3.7$ |
| **Characters** [15] | $71.50 \pm 3.7$ | $96.20 \pm 1.6$ | $97.83 \pm 1.5$ | $96.40 \pm 1.5$ | $97.10 \pm 1.6$ | $\textbf{98.30} \pm 1.4$ | $96.50 \pm 2.1$ | $98.07 \pm 1.3$ |
| **OnlineNews** [19] | $55.33 \pm 3.3$ | $\textbf{56.89} \pm 5.3$ | $55.06 \pm 5.1$ | $49.50 \pm 5.3$ | $54.60 \pm 5.1$ | $50.40 \pm 5.1$ | $55.19 \pm 4.5$ | $50.06 \pm 4.9$ |
| **Continents** | $94.82 \pm 1.6$ | $95.46 \pm 1.3$ | $95.86 \pm 1.3$ | $96.03 \pm 1.3$ | $94.81 \pm 1.4$ | $96.85 \pm 1.0$ | $93.51 \pm 1.6$ | $\textbf{97.18} \pm 1.0$ |
| **Wall** [20] | $51.26 \pm 2.1$ | $68.36 \pm 1.8$ | $80.58 \pm 1.9$ | $68.45 \pm 2.5$ | $64.14 \pm 1.8$ | $82.23 \pm 1.5$ | $66.14 \pm 2.1$ | $\textbf{82.97} \pm 1.6$ |
| **Temperature1** | $65.24 \pm 1.3$ | $\textbf{69.40} \pm 1.3$ | $65.41 \pm 5.2$ | $65.53 \pm 4.0$ | $67.23 \pm 2.4$ | $69.33 \pm 1.2$ | $65.73 \pm 1.2$ | $68.05 \pm 2.1$ |
| **Temperature2** | $67.08 \pm 2.8$ | $71.62 \pm 1.3$ | $73.69 \pm 2.1$ | $70.27 \pm 7.7$ | $67.95 \pm 3.0$ | $76.55 \pm 1.0$ | $76.51 \pm 1.2$ | $\textbf{79.83} \pm 1.3$ |
| **MAGIC** [17] | $73.00 \pm 0.6$ | $80.78 \pm 0.7$ | $\textbf{81.75} \pm 1.3$ | $79.66 \pm 0.9$ | $78.77 \pm 0.7$ | $80.27 \pm 0.8$ | $78.74 \pm 0.7$ | $80.26 \pm 0.8$ |

**Table 2.** Average macro $F_1$-score and standard deviation in the same setting as described in Table 1. Best results are given in bold.

| Dataset | vMNB | dLR | dRBF-SVM | cos-SVM | t-SVM | T-SVM | a-SVM | A-SVM |
|---|---|---|---|---|---|---|---|---|
| **Colposcopy** | $68.96 \pm 11.1$ | $79.61 \pm 10.0$ | $83.32 \pm 8.9$ | $81.21 \pm 7.8$ | $82.86 \pm 9.6$ | $83.32 \pm 8.6$ | $\textbf{85.17} \pm 8.6$ | $82.33 \pm 9.8$ |
| **Behavior** [14] | $22.13 \pm 6.2$ | $51.29 \pm 16.9$ | $42.17 \pm 9.5$ | $43.17 \pm 8.7$ | $\textbf{53.29} \pm 16.1$ | $45.04 \pm 9.7$ | $50.63 \pm 14.3$ | $43.38 \pm 10.7$ |
| **Arrhythmia** [13] | $62.40 \pm 6.3$ | $\textbf{77.80} \pm 6.6$ | $77.10 \pm 5.3$ | $77.12 \pm 5.7$ | $77.14 \pm 6.5$ | $75.21 \pm 6.5$ | $77.38 \pm 6.2$ | $75.02 \pm 6.3$ |
| **eBay** [16] | $83.64 \pm 6.0$ | $85.13 \pm 5.1$ | $\textbf{85.77} \pm 5.1$ | $83.64 \pm 4.8$ | $83.05 \pm 6.5$ | $81.92 \pm 5.8$ | $83.26 \pm 6.6$ | $81.07 \pm 5.3$ |
| **Megaspores** [18] | $\textbf{74.29} \pm 4.0$ | $72.75 \pm 5.4$ | $71.02 \pm 5.9$ | $69.94 \pm 8.1$ | $71.93 \pm 5.4$ | $69.31 \pm 5.9$ | $71.63 \pm 5.0$ | $71.50 \pm 4.8$ |
| **Characters** [15] | $71.73 \pm 4.1$ | $96.13 \pm 1.7$ | $97.82 \pm 1.5$ | $96.33 \pm 1.5$ | $97.09 \pm 1.6$ | $\textbf{98.29} \pm 1.4$ | $96.50 \pm 2.1$ | $98.05 \pm 1.3$ |
| **OnlineNews** [19] | $45.27 \pm 4.9$ | $\textbf{55.36} \pm 5.4$ | $51.89 \pm 6.0$ | $47.94 \pm 5.8$ | $52.74 \pm 5.7$ | $48.86 \pm 5.6$ | $53.50 \pm 5.1$ | $48.30 \pm 5.7$ |
| **Continents** | $93.02 \pm 2.8$ | $94.57 \pm 1.8$ | $95.40 \pm 1.6$ | $95.76 \pm 1.5$ | $91.81 \pm 5.4$ | $96.99 \pm 1.1$ | $88.73 \pm 3.8$ | $\textbf{97.34} \pm 0.9$ |
| **Wall** [20] | $55.31 \pm 2.2$ | $71.66 \pm 2.1$ | $83.57 \pm 1.7$ | $72.16 \pm 2.2$ | $64.12 \pm 2.2$ | $84.60 \pm 1.5$ | $68.06 \pm 3.3$ | $\textbf{85.14} \pm 1.7$ |
| **Temperature1** | $\textbf{59.13} \pm 1.3$ | $48.85 \pm 0.9$ | $46.02 \pm 3.6$ | $46.21 \pm 2.8$ | $47.27 \pm 1.9$ | $48.83 \pm 0.8$ | $46.18 \pm 0.9$ | $47.93 \pm 1.6$ |
| **Temperature2** | $59.39 \pm 5.0$ | $49.64 \pm 1.0$ | $52.27 \pm 3.3$ | $51.81 \pm 3.5$ | $55.94 \pm 4.0$ | $53.91 \pm 0.7$ | $54.53 \pm 3.1$ | $\textbf{74.21} \pm 2.0$ |
| **MAGIC** [17] | $65.79 \pm 0.7$ | $78.28 \pm 0.8$ | $\textbf{78.75} \pm 1.5$ | $77.13 \pm 1.4$ | $75.37 \pm 0.9$ | $77.37 \pm 0.9$ | $75.26 \pm 0.8$ | $77.43 \pm 0.9$ |

Hereafter, we will denote by non-Kernelized directional SVMs (nK-dSVM) the subset of proposed SVM variations with VC dimension equivalent to the one induced by the directional logistic regression; namely, the primal fixed-margin directional SVM with triangle (symmetric and asymmetric) and cosine waves. The remaining models (i.e., directional RBF, symmetric, and asymmetric kernels) will be referred to as Kernelized directional SVMs (K-dSVM).

Although some datasets used here are considerably imbalanced, accuracy and macro $F_1$-score values were fairly consistent with each other, in the sense that the best model in terms of accuracy was the top-1 model in terms of macro $F_1$-score in 10 of 12 datasets and was among the top-2 models in all datasets. While the dLR achieved a competitive general performance, it was surpassed by at least one of the proposed SVM alternatives in most cases. nK-dSVM performed better than dLR on small datasets, given the margin regularization imposed by the SVM loss function. For larger datasets, dLR performed better since the generalization induced by the nK-dSVM margin became less relevant. However, for large datasets, K-dSVM surpassed dLR and their non-kernelized counterparts in most cases. In general, dSVM with asymmetric margins (kernelized and non-kernelized) attained the best results, obtaining the best average performance on half of the datasets.

As shown in Section 5, kernels involving the triangle wave correspond to inner products in an infinite-dimensional feature space. The same is also true for the directional RBF kernel. Non-kernelized methods, on the other hand, are constructed by explicitly defining the feature transformation, having a necessarily finite VC dimension. Therefore, the former produce models with higher capacity, which may lead to overfitting in small datasets, but better accuracy for large ones. This is confirmed by our experiments: the non-kernelized models achieved the best results in small datasets, while kernelized models built on top of the triangle wave and directional RBF kernel attained the best results in large datasets. The performance gains of kernelized models on the larger datasets were small, however, which may be explained by the unimodal distribution of the angular variables. On datasets with a multi-modal distribution of the directional variables, it is expected to observe higher gains by K-dSVM.

*Towards Deep Directional Classifiers*

Deep neural networks have achieved remarkable results in multiple machine learning problems and, particularly, in supervised classification. SVMs, on the one hand, typically decouple the data representation problem from the learning problem, by first projecting the data into a prespecified feature space and then learning a hyperplane that separates the two classes. Deep networks, on the other hand, jointly learn the data representation and the decision function, exhibiting superior performance mostly when trained on large datasets. In the context of directional data, we argue that significant performance improvements might be attained by combining the angular awareness of directional feature transformations or kernels with the representation learning provided by deep neural networks.

In order to evaluate the potential of deep classifiers for directional data, we present two further experiments in this section. Specifically, we have trained two Multilayer Perceptrons (MLPs), which were essentially identical, except for one important difference: one of them (denoted by rMLP) was trained on top of raw angle values (normalized to lie in a single period); the other one (denoted by dMLP) was trained on top of the feature transformation $\phi(x) = (\cos(x), \sin(x))^\intercal$, which defines the cosine kernel, applied to all angular components. The latter was a first attempt towards deep directional classifiers, while the former was completely unaware of the directionality of the data. Each hidden layer in the MLPs had the following structure: fully-connected transformation (dense layer) with 256 output neurons + batch normalization [21] + ReLU + dropout [22]. The output layer is a standard fully-connected transformation followed by a sigmoid, in the case of binary classification, or a softmax, when there are more than two classes. The models were trained to minimize the usual cross-entropy loss with $\ell^2$ regularization. The total number of layers

was chosen between 4 and 5 using 3-fold cross-validation, together with the remaining hyperparameters (dropout rate, $\ell^2$ regularization weight, and learning rate). Training was performed for 200 epochs or until the loss plateaued. The training protocol, including the evaluated datasets, the number of runs for each dataset, and the evaluated metrics, was exactly the same as in the previous set of experiments.

Results are in Table 3a,b, where we show again the values of our most accurate SVM (denoted by best-SVM) in each dataset for easier comparison. Like before, we observed high consistency between the accuracy and macro $F_1$-score values. As expected, rMLP had the worst overall performance, and this effect was mostly apparent in small datasets where the number of directional features was in the same order of magnitude as the number of non-directional ones, like Colposcopy and eBay (see Appendix B). In larger datasets and in those where the number of non-directional features was much larger than the number of directional ones (e.g., Behavior, OnlineNews), rMLP achieved more competitive results. The exception was the MAGIC dataset, where the single directional feature seemed to have a high discriminative power, and so, rMLP achieved the lowest performance among the three models. Contrary to what we just observed for rMLP, the gains of dMLP were highly encouraging. This model, built on top of a directional feature transformation, generally outperformed best-SVM in larger datasets and achieved competitive results even in smaller ones. This observation reinforces the role of directionality in these datasets and, more importantly, motivates the importance of further research to merge directional feature transformations and/or kernels with deep neural networks, which we plan to develop as future work.

**Table 3.** Average performance and standard deviation per model on 30 runs for 12 datasets, which are presented in increasing order of their respective cardinality. The evaluated models are: directional SVM with highest average accuracy (best-SVM), MLP trained on top of raw angle values (rMLP), and MLP trained on top of the feature transformation $\phi(x) = (\cos(x), \sin(x))^\mathsf{T}$ (dMLP). Best results are given in bold.

| (a) Accuracy | | | |
|---|---|---|---|
| **Dataset** | **best-SVM** | **rMLP** | **dMLP** |
| **Colposcopy** | **85.41** $\pm$ 8.5 | 51.67 $\pm$ 18.1 | 81.26 $\pm$ 9.9 |
| **Behavior** [14] | 82.99 $\pm$ 3.8 | 81.85 $\pm$ 5.1 | **83.50** $\pm$ 5.3 |
| **Arrhythmia** [13] | **78.59** $\pm$ 5.2 | 70.17 $\pm$ 9.6 | 71.45 $\pm$ 5.1 |
| **eBay** [16] | **86.00** $\pm$ 5.1 | 73.38 $\pm$ 21.7 | 84.86 $\pm$ 5.3 |
| **Megaspores** [18] | **76.28** $\pm$ 3.7 | 76.01 $\pm$ 4.0 | 73.78 $\pm$ 4.4 |
| **Characters** [15] | **98.30** $\pm$ 1.4 | 83.63 $\pm$ 7.2 | 92.07 $\pm$ 2.7 |
| **OnlineNews** [19] | 50.06 $\pm$ 4.9 | 52.09 $\pm$ 4.1 | **52.33** $\pm$ 4.2 |
| **Continents** | **97.18** $\pm$ 1.0 | 95.57 $\pm$ 1.6 | 96.56 $\pm$ 1.7 |
| **Wall** [20] | 82.97 $\pm$ 1.6 | 86.23 $\pm$ 1.2 | **87.03** $\pm$ 1.7 |
| **Temperature1** | 69.33 $\pm$ 1.2 | 70.02 $\pm$ 2.6 | **70.03** $\pm$ 2.0 |
| **Temperature2** | 79.83 $\pm$ 1.3 | **81.06** $\pm$ 2.5 | 75.63 $\pm$ 4.6 |
| **MAGIC** [17] | 81.75 $\pm$ 1.3 | 73.20 $\pm$ 18.5 | **87.23** $\pm$ 1.3 |

| (b) Macro $F_1$-score | | | |
|---|---|---|---|
| **Dataset** | **best-SVM** | **rMLP** | **dMLP** |
| **Colposcopy** | **85.17** $\pm$ 8.6 | 46.36 $\pm$ 20.2 | 80.86 $\pm$ 10.2 |
| **Behavior** [14] | 53.29 $\pm$ 16.1 | 42.30 $\pm$ 13.6 | **58.03** $\pm$ 15.3 |
| **Arrhythmia** [13] | **77.12** $\pm$ 5.7 | 68.87 $\pm$ 10.0 | 69.76 $\pm$ 5.5 |
| **eBay** [16] | **85.77** $\pm$ 5.1 | 72.30 $\pm$ 23.2 | 84.47 $\pm$ 5.4 |
| **Megaspores** [18] | **71.63** $\pm$ 5.0 | 70.45 $\pm$ 5.7 | 68.61 $\pm$ 5.4 |
| **Characters** [15] | **98.29** $\pm$ 1.4 | 82.56 $\pm$ 8.0 | 91.83 $\pm$ 2.9 |
| **OnlineNews** [19] | 48.30 $\pm$ 5.7 | 49.00 $\pm$ 5.0 | **49.77** $\pm$ 5.8 |
| **Continents** | **97.34** $\pm$ 0.9 | 94.81 $\pm$ 2.3 | 96.30 $\pm$ 2.4 |
| **Wall** [20] | 85.14 $\pm$ 1.7 | 86.53 $\pm$ 1.3 | **88.11** $\pm$ 1.8 |
| **Temperature1** | 48.83 $\pm$ 0.8 | 56.61 $\pm$ 7.4 | **60.91** $\pm$ 5.2 |
| **Temperature2** | 74.21 $\pm$ 2.0 | **74.83** $\pm$ 4.8 | 67.68 $\pm$ 5.0 |
| **MAGIC** [17] | 78.75 $\pm$ 1.5 | 69.16 $\pm$ 21.3 | **85.47** $\pm$ 1.3 |

## 10. Conclusions

Several concepts in real-life applications are represented by directional variables; from periodic time representation on calendars to compass directions. Traditional classifiers, which are unaware of the angular nature of these variables, might not properly model the data. Thereby, the study of directional classifiers is relevant for the machine learning community. Previous attempts to address classification tasks with directional variables focused on generative models [8] and discriminative linear models (logistic regression) [4].

In this work, we proposed several instantiations of directional-aware support vector machines. First, we modified the SVM decision function by considering parametric periodic mappings of the directional variables using cosine and triangle waves. Then, we proposed an extension of the model with triangular waves in order to allow asymmetric margins on the circle. The kernelized versions of these models were proposed as well. Furthermore, we analyzed and demonstrated the expressiveness of each proposed alternative.

In the experimental assessment, the relevance of the proposed models was evaluated, being able to achieve competitive results in most datasets. As expected, when compared to other shallow directional classifiers, kernelized models built on top of the triangle wave attained the best results in larger datasets, due to their large expressive power, which we have proven theoretically. One extra experiment combining a directional feature transformation and a deep neural network showed very promising results and clearly motivates further research.

Since the additional parameters involved in our asymmetric SVMs (in both kernelized and non-kernelized versions) have a periodic impact on the decision boundary or are constrained to a specific domain, using gradient-based optimization techniques may result in sub-optimal models. While this problem was circumvented by using fine-tuning from simpler models, there is research room for the design and exploration of optimization techniques specific for these models. Furthermore, deep multiple kernel learning [23] is an unexplored research line in directional data settings that may lead to a unified framework combining directional kernel machines and deep neural networks.

**Author Contributions:** K.F. and J.S.C. motivated the problem and designed the proposed models; D.P. conceived the mathematical proofs in the Appendix and wrote most parts of the paper; D.P. and K.F. conducted the experiments; J.S.C. supervised the work.

**Funding:** This work was partially financed by the ERDF (European Regional Development Fund) through the Operational Programme for Competitiveness and Internationalisation (COMPETE) 2020 Programme and by National Funds through the Portuguese funding agency, FCT (Fundação para a Ciência e a Tecnologia) within project "POCI-01-0145-FEDER-028857" and also by Fundação para a Ciência e a Tecnologia within Ph.D. Grant Numbers SFRH/BD/93012/2013 and SFRH/BD/129600/2017.

**Conflicts of Interest:** The authors declare no conflict of interest.

## Abbreviations

The following abbreviations are used in this manuscript:

| | |
|---|---|
| dLR | directional Logistic Regression |
| MLP | Multilayer Perceptron |
| SVM | Support Vector Machine |
| vMNB | von-Mises Naive Bayes |
| dSVM | directional SVM |
| K-dSVM | Kernelized directional SVM |
| nK-dSVM | non-Kernelized directional SVM |
| cos-SVM | SVM with cosine kernel |

dRBF-SVM    SVM with directional RBF kernel
a-SVM         primal SVM with asymmetric triangle wave
A-SVM        SVM with Asymmetric triangle wave kernel
t-SVM          primal SVM with triangle wave
T-SVM        SVM with Triangle wave kernel

## Appendix A. Kernelized Symmetric Directional SVM

*Appendix A.1. Kernel Validity: Proof of Theorem 1*

From the definition of $g$, we may verify that it is an even function:

$$g(-x) = \int_0^T h(t)h(t-x)\,dt = \int_{-x}^{T-x} h(x+\tau)h(\tau)\,d\tau = g(x). \tag{A1}$$

Because $g$ is also periodic with period $T$, it may be expressed in a Fourier series of the form:

$$g(x) = \sum_{n=0}^{\infty} a_n \cos\left(\frac{2n\pi x}{T}\right). \tag{A2}$$

Thus,

$$\begin{aligned}
g(x-z) &= \sum_{n=0}^{\infty} a_n \cos\left(\frac{2n\pi(x-z)}{T}\right) \\
&= \sum_{n=0}^{\infty} a_n \cos\left(\frac{2n\pi x}{T}\right)\cos\left(\frac{2n\pi z}{T}\right) + \\
&\quad + \sum_{n=0}^{\infty} a_n \sin\left(\frac{2n\pi x}{T}\right)\sin\left(\frac{2n\pi z}{T}\right) \\
&= \boldsymbol{\phi}(x)^{\mathsf{T}}\boldsymbol{\phi}(z),
\end{aligned} \tag{A3}$$

where:

$$\begin{aligned}
\boldsymbol{\phi}(\cdot) = \bigg(&\sqrt{a_0}, \sqrt{a_1}\cos\left(\frac{2\pi\cdot}{T}\right), \sqrt{a_1}\sin\left(\frac{2\pi\cdot}{T}\right), ..., \\
&\sqrt{a_n}\cos\left(\frac{2n\pi\cdot}{T}\right), \sqrt{a_n}\sin\left(\frac{2n\pi\cdot}{T}\right), ...\bigg)^{\mathsf{T}}
\end{aligned} \tag{A4}$$

Therefore, if $\boldsymbol{\phi}(x)$ and $\boldsymbol{\phi}(z)$ are real vectors, the product $\boldsymbol{\phi}(x)^{\mathsf{T}}\boldsymbol{\phi}(z)$ is an inner product, and so, $g(x-z)$ is a kernel function. Clearly, $\boldsymbol{\phi}(x)$ and $\boldsymbol{\phi}(z)$ are real vectors if and only if $a_n \geq 0$, $\forall\, n$, which can be proven to be true, concluding the proof:

$$\begin{aligned}
a_0 &= \frac{2}{T}\int_0^T g(x)\,dx \\
&= \frac{2}{T}\int_0^T \int_0^T h(t)h(t+x)\,dt\,dx \\
&= \frac{2}{T}\int_0^T h(t)\int_t^{t+T} h(\tau)\,d\tau\,dt \\
&= \frac{2}{T}\left(\int_0^T h(t)\,dt\right)^2 \geq 0,
\end{aligned} \tag{A5}$$

and, for $n \geq 1$,

$$
\begin{aligned}
a_n &= \frac{2}{T} \int_0^T g(x) \cos\left(\frac{2n\pi x}{T}\right) dx \\
&= \frac{2}{T} \int_0^T \int_0^T h(t) h(t+x) \cos\left(\frac{2n\pi x}{T}\right) dt\, dx \\
&= \frac{2}{T} \int_0^T \int_0^T h(t) h(t+x) \cos\left(\frac{2n\pi(t+x)}{T}\right) \cos\left(\frac{2n\pi t}{T}\right) dt\, dx \\
&\quad + \frac{2}{T} \int_0^T \int_0^T h(t) h(t+x) \sin\left(\frac{2n\pi(t+x)}{T}\right) \sin\left(\frac{2n\pi t}{T}\right) dt\, dx \\
&= \frac{2}{T} \int_0^T h(t) \cos\left(\frac{2n\pi t}{T}\right) \int_t^{t+T} h(\tau) \cos\left(\frac{2n\pi\tau}{T}\right) d\tau\, dt \\
&\quad + \frac{2}{T} \int_0^T h(t) \sin\left(\frac{2n\pi t}{T}\right) \int_t^{t+T} h(\tau) \sin\left(\frac{2n\pi\tau}{T}\right) d\tau\, dt \\
&= \frac{2}{T} \left( \int_0^T h(t) \cos\left(\frac{2n\pi t}{T}\right) dt \right)^2 \\
&\quad + \frac{2}{T} \left( \int_0^T h(t) \sin\left(\frac{2n\pi t}{T}\right) dt \right)^2 \geq 0.
\end{aligned}
\tag{A6}
$$

*Appendix A.2. VC Dimension: Proof of Theorem 2*

Before going into the details of the proof of Theorem 2, we need the result presented in Lemma A1.

**Lemma A1.** *Let* $\mathbb{F} = \{\phi_1, \cdots, \phi_N\}$ *be a set of square integrable and non-zero functions* $\phi_i : \mathbb{I} \subseteq \mathbb{R} \mapsto \mathbb{R}$, *where* $\mathbb{I}$ *is an interval, that satisfy:*

$$
\int_{\mathbb{I}} \phi_i(x)^2 \neq 0, \, \forall i, \quad \int_{\mathbb{I}} \phi_i(x)\phi_j(x) = 0, \, \forall i \neq j.
\tag{A7}
$$

*There exists a vector* $x = (x_1, ..., x_N)^\intercal \in \mathbb{I}^N$ *such that the matrix:*

$$
\Phi(x) = \begin{pmatrix} \phi_1(x_1) & \cdots & \phi_N(x_1) \\ \vdots & \ddots & \vdots \\ \phi_1(x_N) & \cdots & \phi_N(x_N) \end{pmatrix}
\tag{A8}
$$

*has full rank.*

**Proof.** We shall prove by contradiction that such $x$ actually exists. Suppose that $\Phi(x)$ is rank deficient for all $x$. Then, there exists a function $v : \mathbb{I}^N \mapsto \mathbb{R}^N$ such that:

$$
\Phi(x)v(x) = \mathbf{0} \text{ and } v(x) \neq \mathbf{0}, \, \forall x.
\tag{A9}
$$

Due to the orthogonality of the functions in $\mathbb{F}$, no non-trivial linear combination of them vanishes identically in $\mathbb{I}$, and consequently, $v$ may not be a constant function. However, if $v$ is not constant, there exist distinct vectors $x^{(1)}, ..., x^{(k)}$ with dimension $N$ such that no non-zero vector belongs to the null spaces of all the matrices $\Phi\left(x^{(1)}\right), ..., \Phi\left(x^{(k)}\right)$. This means that the space generated by the rows of these matrices stacked altogether has dimension $N$, so we may choose $N$ linearly-independent rows from this stacked matrix. Since each row is defined by a single element in one of the vectors $x^{(1)}, ..., x^{(k)}$, choosing $N$

linearly-independent rows corresponds to finding a dataset $x^*$ with size $N$ such that $\Phi(x^*)$ has full rank, contradicting the initial hypothesis. $\square$

Now, we may proceed to the proof of Theorem 2. Let $VC(l)$ denote the VC dimension of the class of classifiers $l$. Firstly, we are going to prove that $VC(l) \leq 2N + 1$. If we proceed as in the proof of Theorem 1, $\phi$ may be defined as a feature space over $\mathbb{R}^{2N}$, by suppressing all components whose coefficients are zero. Therefore, $f$ becomes a hyperplane in $\mathbb{R}^{2N}$, and so, it cannot shatter more than $2N + 1$ data points.

Now, it suffices to prove that $VC(l) \geq 2N + 1$. Let us denote the period of $g$ by $T$ and consider a dataset with $\bar{N} = 2N$ examples, namely $x = (x_1, ..., x_{\bar{N}})^\mathsf{T}$, where $0 \leq x_i < T$, $\forall i$. Like before, we obtain $\phi$ as in the proof of Theorem 1, and we build the $\bar{N} \times \bar{N}$ matrix $\Phi$, defined as:

$$\Phi(x) = \begin{pmatrix} \phi(x_1)^\mathsf{T} \\ \vdots \\ \phi(x_{\bar{N}})^\mathsf{T} \end{pmatrix}. \tag{A10}$$

By further defining $f(x|w_0, w_d) = (f(x_1|w_0, w_d), ..., f(x_{\bar{N}}|w_0, w_d))^\mathsf{T}$, the following equality is straightforward to check:

$$f(x|w_0, w_d) = w_0 \oplus \Phi(x)w_d, \tag{A11}$$

where $\oplus$ denotes the operation of summing the scalar on the left-hand side to every element of the vector on the right-hand side. Let us denote the ground truth label of each $x_i$ as $y_i \in \{-1, 1\}$. Because the elements of $\phi(x)$ form a set of orthogonal functions, we know from Lemma A1 that there exists a dataset $x$ such that $\Phi(x)$ has full rank. Thus, from now on, assume this condition holds. This assumption ensures that, for all possible combinations of $y_i$ values, there exists a $w_d \in \mathbb{R}^{\bar{N}}$ that satisfies:

$$\Phi(x)w_d = \begin{pmatrix} y_1 \\ \vdots \\ y_{\bar{N}} \end{pmatrix}. \tag{A12}$$

Clearly, using this $w_d$, the classifier labels all $\bar{N}$ data points correctly provided that $|w_0| < 1$. Now, suppose we have one more data point $x_{\bar{N}+1} \in [0, T)$, with ground truth label $y_{\bar{N}+1} \in \{-1, 1\}$. Assume, furthermore, that $x_{\bar{N}+1}$ is such that $|w_d^\mathsf{T} \phi(x_{\bar{N}+1})| < 1$. The existence of such $x_{\bar{N}+1}$ is guaranteed, since the elements of $\phi$ are continuous functions with zero mean value. By setting $w_0 = y_{\bar{N}+1}\delta - w_d^\mathsf{T}\phi(x_{\bar{N}+1})$, for any arbitrary $0 < \delta < 1 - |w_d^\mathsf{T}\phi(x_{\bar{N}+1})|$, the classifier labels all $\bar{N} + 1$ data points correctly. Thus, $VC(l) \geq \bar{N} + 1 = 2N + 1$.

## Appendix B. Summary of the Datasets

We give a summary of the main characteristics of the datasets used in this work, including number of features per type (i.e., Directional (Dir), Linear (Lin), Discrete (Disc)) and the number of samples per dataset (#).

**Table A1.** Number of variables, number of classes and cardinality of each dataset. Variables are divided into the respective types: directional (Dir), linear (Lin) and discrete (Disc).

| Dataset | Number of Variables | | | Class Values | # |
|---|---|---|---|---|---|
| | Dir | Lin | Disc | | |
| **Colposcopy** | 3 | 6 | 0 | 3 | 150 |
| **Behavior** [14] | 140 | 426 | 20 | 4 | 261 |
| **Arrhythmia** [13] | 4 | 191 | 66 | 2 | 430 |
| **eBay** [16] | 1 | 2 | 0 | 11 | 528 |
| **Megaspores** [18] | 1 | 0 | 0 | 2 | 960 |
| **Characters** [15] | 5 | 31 | 0 | 10 | 1000 |
| **OnlineNews** [19] | 1 | 12 | 0 | 2 | 1000 |
| **Continents** | 2 | 0 | 0 | 5 | 3481 |
| **Wall** [20] | 6 | 6 | 0 | 4 | 5456 |
| **Temperature1** | 2 | 1 | 0 | 3 | 8764 |
| **Temperature2** | 5 | 1 | 0 | 3 | 8764 |
| **MAGIC** [17] | 1 | 10 | 0 | 2 | 19,020 |

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
