# Peer review of "Directional Support Vector Machines"

_applsci, doi:10.3390/app9040725_

Round 1

Reviewer 1 Report

This paper proposes several variants of Directional Support Vector Machines. Experimental results show that the proposed method can obtain competitive performance against the directional naive Bayes and directional Logistic Regression algorithms.

My major concern about this paper is the motivation. As deep learning becomes more and more popular, how would the proposed method be compared with deep learning approach in theory and experiments? 

Some related reference should be included, e.g., Jian Wang, Feng Zhou, Shilei Wen, Xiao Liu, and Yuanqing Lin. "Deep metric learning with angular loss." In IEEE International Conference on Computer Vision (ICCV), pp. 2612-2620, 2017.

More experiments should be performed. For example, the proposed method should be compared with SVM based on different loss functions as well as state-of-the-art classifiers, e.g., deep classifiers.

In Table 1, average accuracy/ranking performance should be reported.

It is mentioned that directional/angular/periodic property is important. However, I am sure whether such property is valid on the datasets used for experiments.

Author Response

Please find our point-by-point response to the reviewer’s comments in the attached PDF file.

Reviewer 2 Report

DIRECTIONAL SUPPORT VECTOR MACHINES

The paper proposes directional support vector machines for the analysis of directional data against the von Mises naive Bayes and the directional Logistic Regression. The paper includes some theoretical and experimental results.

I have the following observations:

1)      When comparing different algorithms only accuracy is considered, in my opinion other metrics like sensivity, specificity, precision, etc should be included.

2)      In line 55, please U stands for?.

3)      In line 107 “kernelized version 11”? is that correct?.

4)      Is Figure 2 referenced in the text?

5)      Are references from 13 to 20 referenced in the text.

6)      I would prefer a more extended introduction, however I leave to the authors this decisión.

Author Response

(The authors gave the same response as above.)

Round 2

Reviewer 1 Report

The quality of this manuscript has been improved. I would recommend acceptance of this paper.

Author Response

Dear Reviewer,

Thank you once again for reviewing our paper. We are glad that you appreciated the new content that we have added to the manuscript and that you now recommend its acceptance for publication in this journal.

We have made two minor changes to our manuscript:

Following a suggestion from another reviewer, we have added references 15-22 to the text (lines 184 and 185). These references are for the datasets that we have used and were formerly present in the tables only.

We have disclosed the URL of the source code repository, which is in the footnote on page 9.

Reviewer 2 Report

The article has been improved. However, the references in the tables are a bit strange, in my opinion the authors should include a comment in the text that includes the references.

Author Response

Dear Reviewer,

Thank you once again for reviewing your paper. We are glad that you appreciated the new content that we have added to the manuscript

«in my opinion the authors should include a comment in the text that includes the references.»

We have just added an enumeration of the evaluated datasets, together with the respective references (lines 184 and 185). We decided to keep them in the tables also, for easier reference. 

We have also disclosed the URL of the source code repository, which is in the footnote of page 9.